# Arterial Stiffness in Thyroid and Parathyroid Disease: A Review of Clinical Studies

**DOI:** 10.3390/jcm11113146

**Published:** 2022-06-01

**Authors:** Andrea Grillo, Vincenzo Barbato, Roberta Maria Antonello, Marco Fabio Cola, Gianfranco Parati, Paolo Salvi, Bruno Fabris, Stella Bernardi

**Affiliations:** 1Department of Medicine, Surgery and Health Sciences, University of Trieste, 34149 Trieste, Italy; barbato.vinc@gmail.com (V.B.); rma.roby@gmail.com (R.M.A.); marcofabio.cola@asugi.sanita.fvg.it (M.F.C.); b.fabris@fmc.units.it (B.F.); stella.bernardi@asugi.sanita.fvg.it (S.B.); 2SC Medicina Clinica, ASUGI (Azienda Sanitaria Universitaria Giuliano Isontina), Cattinara Hospital, 34149 Trieste, Italy; 3Department of Cardiology, Istituto Auxologico Italiano, IRCCS, 20122 Milan, Italy; gianfranco.parati@unimib.it (G.P.); psalvi.md@gmail.com (P.S.); 4Department of Medicine and Surgery, University of Milano-Bicocca, 20126 Milan, Italy

**Keywords:** arterial stiffness, thyroid, parathyroid, cardiovascular disease

## Abstract

Growing evidence shows that arterial stiffness measurement provides important prognostic information and improves clinical stratification of cardiovascular risk. Thyroid and parathyroid diseases are endocrine diseases with a relevant cardiovascular burden. The objective of this review was to consider the relationship between arterial stiffness and thyroid and parathyroid diseases in human clinical studies. We performed a systematic literature review of articles published in PubMed/MEDLINE from inception to December 2021, restricted to English languages and to human adults. We selected relevant articles about the relationship between arterial stiffness and thyroid and parathyroid diseases. For each selected article, data on arterial stiffness were extracted and factors that may have an impact on arterial stiffness were identified. We considered 24 papers concerning hypothyroidism, 9 hyperthyroidism and 16 primary hyperparathyroidism and hypoparathyroidism. Most studies evidenced an increase in arterial stiffness biomarkers in hypothyroidism, hyperthyroidism and primary hyperparathyroidism, even in subclinical and mild forms, although heterogeneity of measurement methods and of study designs prevented a definitive conclusion, suggesting that the assessment of arterial stiffness may be considered in the clinical evaluation of cardiovascular risk in these diseases.

## 1. Introduction

Growing evidence shows that arterial stiffness measurement provides important prognostic information and improves clinical stratification of cardiovascular disease [1,2]. The stiffening of arteries is notably linked to aging, but a number of risk factors and diseases may affect this process, which is usually referred as arteriosclerosis [3]. The biological mechanisms underlying arterial stiffening involve the degradation of elastin layers in the tunica media of the large arteries and the proportional increase in collagen fiber content, along with smooth muscle cell hyperplasia, fibrosis and calcification of the media [4]. The process of stiffening interacts with atheromatous plaque formation and inflammation in the development and progression of cardiovascular disease [5]. The use of arterial stiffness markers is considered of clinical interest to assess cardiovascular risk in particular among patients not presenting the classical risk factors (e.g., smoking, obesity, diabetes), in which evaluation of arterial damage may improve risk stratification [6].

Thyroid and parathyroid diseases should be considered as endocrine diseases with relevant implications in the cardiovascular system. Patients with overt hyperthyroidism or hypothyroidism show several alterations, caused either by effects of thyroid hormones in the heart and in the vasculature or by cardiovascular risk factors (including blood pressure, dyslipidemia and inflammation), which may increase cardiovascular risk and lead to cardiovascular morbidity and mortality [7,8]. Similarly, patients with primary hyperparathyroidism often present cardiovascular abnormalities and an increased cardiovascular risk [9]. Less evidence supports the cardiovascular significance of subclinical hyper- or hypothyroidism or mild hyperparathyroidism, although detrimental cardiovascular consequences may affect these conditions as well [10,11,12].

In recent years, numerous clinical studies have investigated the relationship between thyroid and parathyroid diseases and arterial stiffness markers. New evidence explains the molecular and physiological pathways leading to vascular disease in conditions of hormonal excess or deficiency. The use of arterial stiffness measures has been proposed to improve clinical management of these conditions in several clinical studies.

In this review, we aimed to perform a literature search for the effects of thyroid and parathyroid hormone excess or deficiency on the structural and functional alterations of the arterial wall which leads to arterial stiffening, focusing on clinical studies conducted in humans.

## 2. Material and methods

A PubMed/MEDLINE search was performed to select peer-reviewed articles published from inception to 31 December 2021. The complete search string included inclusive keywords regarding arterial stiffness (e.g., “arterial stiffness”, “arterial compliance”, “pulse wave velocity”, “PWV”, “augmentation index”, “AIx”) and inclusive keywords regarding the evaluated hormones and their imbalance (“hypothyroidism” or “thyroid hormones” or “thyrotoxicosis” or “levothyroxine” or “Hashimoto” or “Graves” or “Basedow” or “dysthyroidism” “hyperparathyroidism” or “PTH”). Papers written in languages other than English, not pertinent with the present review, or whose full text was not available were excluded. The pertinent references were evaluated and eventually included in the final manuscript. We considered all papers in open-access and non-open access journals. The included papers were summarized and their results discussed in the text, offering an overview of current literature. Clinical studies in adult humans were organized in Tables (Table 1: hypothyroidism, Table 2: hyperthyroidism, Table 3: primary hyperparathyroidism). The manuscript was organized in the following major chapters: (1) Thyroid: hypo- and hyper-thyroidism; (2) Parathyroid.

## 3. Results

### 3.1. Thyroid

Our literature search identified 33 papers regarding dysthyroidism and its impact on arterial stiffness. Twenty-five studies evaluated arterial stiffness in patients with overt and/or subclinical hypothyroidism, while nine studies evaluated patients with overt and/or subclinical hyperthyroidism (one study evaluated patients with Hashimoto thyroiditis including both hyperthyroid and hypothyroid patients). From an overall evaluation, variable results about the impact of thyroid hormones on arterial stiffness are reported. In addition, arterial stiffness was assessed using different measurement methods, varying from ultrasounds to tonometry. Considered measures of arterial stiffness included pulse wave velocity (PWV), augmentation index and other surrogate measures. PWV is the velocity at which the pressure pulsation propagates through the circulatory system, and is defined as the distance travelled by the pulse wave divided by the time [2], and the carotid-femoral PWV is the gold standard for measuring stiffness of the aorta [3]. Variability in methodologies, together with the differences in study populations and in inclusion/exclusion criteria among studies, may explain the heterogeneity of results.

#### 3.1.1. Overt and Subclinical Hypothyroidism

The prevalence of subclinical hypothyroidism, defined as a serum thyroid stimulating hormone (TSH) level above the upper normal limit despite normal levels of serum free thyroxine (fT4), in the general population is 5–15% [13]. Of these, 2–5% progress to overt hypothyroidism [14]. Clinical and subclinical hypothyroidism are related to cardiovascular diseases, in particular atherosclerosis and ischemic heart disease [15]. Over the past years, the hypothesis that even in euthyroid subjects thyroid function may affect cardiovascular health has been supported by different authors [16,17,18]. A large Chinese population-based, cross-sectional study, including 812 euthyroid subjects without major cardiovascular risk factors, showed a significant and independent association of fT4 with baPWV in euthyroid subjects [19].

Regarding subclinical hypothyroidism, evidence observing an increase in arterial stiffness is conflicting and heterogeneous, both in methods used to evaluate arterial stiffness and in results. A number of small studies were conducted to evaluate arterial stiffness in subclinical hypothyroidism, in small cohorts of patients and with different methodologies. Earlier studies mainly considered the evaluation of brachial-ankle PWV, which was found to be increased [20] but not correlated to TSH [21]. Further studies confirmed the increase in aortic stiffness, measured either with brachial-ankle or heart-femoral PWV [21,22] and observed a decrease in stiffness after l-thyroxine replacement [23,24]. The increase in stiffness was confirmed by other methodologies, as the evaluation of β-stiffness index in the carotid artery [25,26,27], or the augmentation index [28,29], while the evaluation of global aortic distensibility gave contrasting results [30,31]. Considering the response to treatment, the augmentation index tended to reduce with levothyroxine replacement therapy [32].

More recent studies explored the relationship between the PWV in aorta and subclinical hypothyroidism. Three studies found an increase in aortic PWV in this condition [33,34,35], although one study conducted in a large sample and with gold-standard measurement (carotid-femoral PWV) [36] did not find an association between subclinical hypothyroidism and aortic stiffness. The CAVI, a blood pressure-independent stiffness index of the aorta, which is related to central and peripheral arterial stiffness and to 24 h blood pressure [37], was found to be increased in subclinical hypothyroidism [38], with a possible benefit after acute aerobic exercise [39].

Similarly to subclinical, most studies conducted in overt hypothyroidism found an increase in arterial stiffness, which has been evaluated across studies with a variety of biomarkers. Earlier studies evaluated markers related to reflected waves, as augmentation index [40,41]. Timing and magnitude of reflected waves was determined by hypothyroidism and improved after replacement therapy, indicating a positive effect on arterial stiffness [40]. Other studies focused on biomarkers of structural arterial stiffening, as β index [25,42,43], pulsatility index in carotid arteries [44] or brachial-ankle PWV [21]. One large study conducted in a large population, with brachial-ankle PWV, did not find a significant difference between hypothyroid and euthyroid subjects, which adds heterogeneity in the results [45]. More recently, studies that evaluated the gold-standard measure of aortic stiffness, the carotid-femoral PWV [33,44] or the heart-femoral PWV [35], found an increase in arterial stiffness markers in patients with overt hypothyroidism.

One study conducted in thyroidectomized patients on long-term replacement therapy, did not find an increase in arterial stiffness, suggesting that targeting TSH in the reference range does not seem to cause adverse cardiovascular effects [46].

**Table 1 jcm-11-03146-t001:** Experimental studies evaluating arterial stiffness and hypothyroidism.

AUTHOR	YEAR, COUNTRY	STUDY DESIGN	DISEASE	OUTCOME MEASURE	STUDY POPULATION	RESULTS
Demellis et al. [42]	2002, Greece	Paired and unpaired case–control study	Hypothyroidism	β index	15 hypothyroid patients, 15 hypertensive patients, 15 hypothyroid and hypertensive patients, 30 controls	Increased aortic stiffness in patients with hypothyroidism and hypertension, reversible by hormone replacement in 50%.
Obuobie et al. [29]	2002, United Kingdom	Paired and unpaired case–control study	Hypothyroidism	AIx, CBP, reflected waves time	12 hypothyroid patients before and after levothyroxine replacement treatment	Hypothyroid patients had significantly higher AIx and CBP and lower reflected waves time than controls; 6 months of levothyroxine replacement therapy reversed the abnormalities.
Dagre et al. [41]	2005, Greece	Cross-sectional study	Hypothyroidism	Augmentation pressure	15 overt hypothyroidism, 50 controls with varying mean TSH serum levels	Serum TSH values were positively correlated with augmentation pressure.
Hamano et al. [21]	2005, Japan	Cross-sectional study	Subclinical hypothyroidism, overt hypothyroidism	baPWV	7 overt hypothyroidism (before and after treatment with L-thyroxin), 28 subclinical hypothyroidism	baPWV was not correlated to TSH. After replacement therapy, fT4 increased and baPWV decreased.
Nagasaki et al. [43]	2005, Japan	Paired and unpaired case–control study	Hypothyroidism before and after levothyroxine replacement therapy	β index, cIMT	30 hypothyroid patients before and after 1 year of treatment	β index higher in hypothyroid patients than controls. After 1 year of treatment significant decreases of β index.
Nagasaki et al. [20]	2006, Japan	Case-control study	Subclinical hypothyroidism	baPWV	50 subclinical hypothyroidism, 50 controls	BaPWV was increased but not correlated with T3, T4 or TSH.
Owen et al. [28]	2006, United Kingdom	Paired and unpaired case–control study	Subclinical hypothyroidism	AIx, CBP, reflected waves time	19 subclinical hypothyroidism (before and after treatment with L-thyroxin), 10 controls	Increased AIx in subclinical hypothyroidism which improved with L-thyroxin
Nagasaki et al. [22]	2007, Japan	Prospective observational study	Subclinical hypothyroidism	hfPWV, faPWV, baPWV	40 subclinical hypothyroidism, 50 controls	hfPWV, faPWV and baPWV were significantly higher in patients with subclinical hypothyroidism compared to controls.
Nagasaki et al. [24]	2007, Japan	Paired case–control study	Subclinical hypothyroidism	baPWV	42 subclinical hypothyroidism, evaluated before and after levothyroxine (L-T(4))	Replacement therapy decreased baPWV.
Nagasaki et al. [25]	2007, Japan	Paired and unpaired case–control study	Hypothyroidism	β index	46 hypothyroid patients (of which 35 evaluated before and after levothyroxine replacement therapy), 46 controls	β index was significantly higher in hypothyroid patients than in controls. After one year of replacement therapy, β index reduced.
Peleg et al. [32]	2008, Israel	Paired case–control study	Subclinical hypothyroidism	AIx	30 patients with subclinical hypothyroidism treated with levothyroxine and assessed at baseline and after 1, 4 and 7 months	The AIx significantly reduced with treatment.
Nagasaki et al. [23]	2009, Japan	Double-blind,	Subclinical hypothyroidism in Hashimoto thyroiditis, before and after levothyroxine replacement treatment	baPWV	95 subclinical hypothyroidism before and after levothyroxine replacement treatment, 42 controls	The baseline baPWV values in patients with subclinical hypothyroidism were significantly higher than in normal subjects. BaPWV showed a significant decrease with treatment. Changes in baPWV and TSH were not correlated.
Tian et al. [26]	2010, China	Case-control study	Subclinical hypothyroidism	β index	93 subclinical hypothyroidism, 90 controls	β index was significantly higher in patients with subclinical hypothyroidism than in normal controls
	2013, Turkey	Case-control study	Subclinical hypothyroidism	ASI, aortic distensibility	43 subclinical hypothyroidism, 48 controls	Aortic distensibility was significantly lower, and ASI was significantly higher in subclinical hypothyroidism than in controls. TSH level was positively correlated with ASI.
Kilic et al. [30]	2013, Turkey	Case-control study	Subclinical hypothyroidism	cIMT, Aortic distensibility	32 subclinical hypothyroidism, 29 controls	No difference in aortic distensibility between subclinical hypothyroidism and controls.
Masaki et al. [38]	2014, Japan	Cross-sectional study	Subclinical Hypothyroidism	CAVI	83 subclinical hypothyroidism, 83 controls	CAVI was increased and associated with high NT-proBNP in subclinical hypothyroidism.
Owecki et al. [44]	2015, Poland	Case-control study	Treated hypothyroidism and euthyroid autoimmune thyroiditis	Pulsatility index in carotid arteries	31 treated hypothyroidism, 26 euthyoroid thytoiditis	Overt hypothyroidism have increased pulsatiliy index in common and internal carotid arteries than euthyroid patients with autoimmune thoiditis
Tudoran [33]	2015, Romania	Paired and unpaired case–control study	Subclinical hypothyroidism, overt hypothyroidism	PWV, AIx	41 overt hypothyroidism, 15 subclinical hypothyroidism, 15 controls. Before and after treatment with L-thyroxin.	All patients had higher PWV and AIx compared with controls. After treatment, PWV and of AIx reduced in the majority of patients.
Feng et al. [27]	2016, China	Cross-sectional study	Hypothyroidism	β index, PWV, distensibility coefficient of carotid arteries	Autoimmune throiditis: 59 hyperthyroidism, 61 hypothyroidism, 60 euthyroidism and controls.	PWV and distensibility coefficients may discrimnate hypo- and hyper-throidism patients.
Laugesen et al. [46]	2016, Demark	Cross-sectional study	Thyroidectomized patients	cfPWV, AIx	30 thyroidectomized patients on long-term replacement therapy	PWV and AIx were not significantly higher in patients compared to controls
Peixoto de Miranda et al. [36]	2017, Brazil	Cross-sectional study	Subclinical hypothyroidism	cfPWV	463 subclinical hypothyroidism, 7878 controls	Subclinical hypothyroidism was not associated with increased cf-PWV.
Masaki et al. [39]	2019, Japan	Prospective observational study	Subclinical hypothyroidism	CAVI	53 subclinical hypothyroidism, 55 controls	The CAVI and serum TSH levels significantly decreased after acute aerobic exercise in the subclinical hypothyroidism group and euthyroid group.
Tanriverdi et al. [34]	2019, Turkey	Cross-sectional study	Subclinical hypothyroidism	PWV	32 subclinical hypothyroidism, 28 controls	PWV was significantly higher in the subclinical hypothyroidism group.

TSH: thyroid stimulating factor; CAVI: cardio-ankle vascular stiffness; PWV: pulse wave velocity; cfPWV: carotid-femoral pulse wave velocity; hfPWV: heart-femoral pulse wave velocity; faPWV: femoral-ankle pulse wave velocity; baPWV: brachial-ankle pulse wave velocity; AIx: augmentation index; ASI: arterial stiffness index; cIMT: carotid intima-media thickness; CBP: central blood pressure.

#### 3.1.2. Overt and Subclinical Hyperthyroidism

Studies conducted in patients with overt hyperthyroidism suggest an increase in structural arterial stiffness biomarkers. An increase in β index in carotid arteries was found, which was reduced by antithyroid drug or radioiodine treatment [47,48]. Additionally, a marker of total arterial compliance (Pulse pressure/stroke volume) was found to be reduced in hyperthyroidism and normalized after beta-blockers therapy [49]. Considering a surrogate biomarker of central arterial stiffness derived from 24 h blood pressure monitoring, the ambulatory arterial stiffness index (AASI), one study did not find a significant difference between patients with overt or subclinical hyperthyroidism [50]. Data on subclinical hyperthyroidism (defined as a low or undetectable TSH, with normal fT3 and fT4) are scarce, considering that only one other work [51] has evaluated patients with this condition after thyroidectomy and l-thyroxine suppressive therapy, finding an increase in the β aortic stiffness by echocardiography.

Two studies evaluated the algorithm-derived PWV calculated from Mobil-O-Graph device in patients with hyperthyroidism, finding an increase in PWV in the office setting [52], which is not surprising considering that the PWV estimated by this method is strictly dependent on actual blood pressure values [53]. Interestingly, in another study the Mobil-O-Graph-derived PWV was not different in hyperthyroidism patients compared to blood-pressure matched controls [54], although the circadian profile of PWV was altered.

The evaluation of markers of reflected waves magnitude and the analysis of central pressure waves gave variable results among studies. In thyrotoxicosis, Obuobie et al. found a decrease in AIx despite an increase in central pulse pressure [40], thus suggesting a lowered central arterial stiffness. Conversely, Bodlaj et al. [55] found an increase in aortic AIx in hyperthyroidism patients with Graves’ disease. A negative correlation between AIx and TSH, and a positive correlation between AIx and free thyroid hormones (fT3, fT4) was also described by Yildiz et al. [52]. Interpretation of these contrasting results probably resides in the close dependence of reflected waves timing with heart rate, which is acutely affected by thyroid hormones [56]. Alterations in functional markers of arterial stiffness can acutely affect the cardiovascular system, producing possible adverse organ damage, but the long-term effects of these transient changes remain to be elucidated.

**Table 2 jcm-11-03146-t002:** Experimental studies evaluating arterial stiffness and hyperthyroidism.

AUTHOR	YEAR, COUNTRY	STUDY DESIGN	DISEASE	OUTCOME MEASURE	STUDY POPULATION	RESULTS
Inaba et al. [47]	2002, Japan	Paired case–control study	Hyperthyroidism (Graves’ disease)	β index (common carotid artery)	27 patients with GD before and after antithyroid drug therapy	Increased β index in hyperthyroidism, reduced by antithyroid drug therapy.
Obuobie et al. [40]	2002, United Kingdom	Paired and unpaired case–control study	Thyrotoxicosis	AIx	20 thyrotoxic patients(before and after treatment with (131)I) and 20 controls	Lower Aix and higher central PP at baseline. Aix reduced at 6 months following treatment with radioiodine therapy
Palmieri et al. [49]	2004, Italy	Paired and unpaired case–control study	Thyrotoxicosis (Graves’ disease)	PP/stroke volume (total arterial stiffness)	20 thyrotoxic patients (before and after treatment with bisoprolol) and 20 controls	Thyrotoxicosis is associated with increased total arterial stiffness. Beta blockade normalized total arterial stiffness.
Bodlaj et al. [55]	2007, Austria	Paired case–control study	Hyperthyroidism (Graves’ disease)	AIx, SEVR	59 patients with Graves’ disease before (hyperthyroidism) and after antithyroid drug treatment (euthyroidism)	AIx was higher and SEVR lower in hyperthyroidism, restored after antithyroid drug (ATD) treatment.
Gazdag et al. [51]	2014, Hungary	Cross-sectional study	Thyroidectomized patients for differentiated thyroid cancer	ASI	24 differentiated thyroid cancer after total thyroidectomy and radioiodine ablation, evaluated on TSH suppressive L-T4 therapy, and 4 weeks after L-T4 withdrawal, 24 controls	Aortic stiffness was increased both in hypothyroidism and subclinical hyperthyroidism compared to controls.
Kang et al. [48]	2015, China	Paired and unpaired case–control study	Hyperthyroidism	β index, PWV, IMT	70 hyperthyroidism before and after (131)I treatment, 74 controls	β index and PWV were higher in patients than in the control group. After treatment, PWV and β were lower than baseline.
İyidir et al. [50]	2017, Turkey	Cross-sectional study	Overt and subclinical hyperthyroidism	AASI	23 overt hyperthyroidism, 36 subclinical hyperthyroidism, 25 controls	AASI did not differ between overt and subclinical hyperthyroidism, but there was a positive relationship between AASI and free thyroid hormone levels.
Yildiz et al. [52]	2019, Turkey	Case-control study	Overt and subclinical hyperthyroidism	PWV, AIx	30 overt hyperthyroid, 28 subclinical hyperthyroid, 14 treated hyperthyroidism, 30 controls	PWV and Aix measurements were significantly higher in the hyperthyroid subclinical hyperthyroid group than in the control group.
Grove-Laugesen et al. [54]	2020, Denmark	Cross-sectional study	Overt hyperthyroidism	PWV (office and 24 h measurement)	55 overt hyperthyroidism (Graves’ disease), 55 controls	Patients with Graves’ disease showed higher PWV in the 24 h but not in the office setting.

TSH: thyroid stimulating factor; AIx: augmentation index; SEVR: subendocardial variability ratio; PP: pulse pressure; AASI: ambulatory arterial stiffness index; ASI: arterial stiffness index; PWV: pulse wave velocity; IMT: intima-media thickness.

### 3.2. Parathyroid

Primary hyperparathyroidism (pHPT) is another very common endocrine disorder, affecting between 0.4% and 11% of the population [57]. The highest rates are due to patients—mostly post-menopausal women—with mild pHPT, who exhibit inappropriately high levels of parathyroid hormone (PTH) and normal or only mildly elevated calcium levels [58]. Parathyroidectomy remains the treatment of choice for this condition, and it is recommended in patients with mild pHPT and normal calcium levels in case of young age, or signs of kidney and/or bone damage [59]. Although the kidney and the bone are the main targets of PTH actions, PTH exerts direct actions on the cardiovascular system too, including not only cardiac myocytes but also endothelial and vascular smooth muscle cells, leading to arterial remodeling [60,61,62].

Based on this background, several works have evaluated the effect of mild pHPT and parathyroidectomy on arterial stiffness. PTH has been found associated with arterial stiffness in the general population [63,64], and mild pHPT has been found associated with an increase of arterial stiffness, as assessed by AIx [65,66] as well as PWV [67,68,69,70,71], additionally to detrimental effects in other forms of vascular organ involvement as aortic intima-media thickness [72] and central blood pressure [73]. By contrast, the studies evaluating the effects of parathyroidectomy upon arterial stiffness have reported conflicting data regarding effects on arterial structure and function [74,75,76,77,78]. Nevertheless, in a recent meta-analysis we found that mild pHPT was associated with an increase of arterial stiffness, which was significantly reduced by parathyroidectomy [79]. These data are in line with the results of another meta-analysis analyzing the effects of parathyroidectomy upon left ventricular mass, where surgery was able to reduce it significantly [80]. These data suggest that surgery could improve arterial stiffness, as well as other signs of cardiovascular organ damage such as left ventricular hypertrophy, and reduce the cardiovascular risk profile in patients with mild pHPT.

Interestingly, hypoparathyroidism has also been associated with an increase of arterial stiffness [81,82], consistent with the higher risk of cardiovascular disease of patients affected by this condition.

**Table 3 jcm-11-03146-t003:** Experimental studies evaluating arterial stiffness and parathyroid disease in humans.

AUTHOR	YEAR, COUNTRY	STUDY DESIGN	DISEASE	OUTCOME MEASURE	STUDY POPULATION	RESULTS
Barletta et al. [74]	2000, Italy	Prospective case–control study	Mild asymptomatic pHPT	PWV IMT	24 patients with mild asymptomatic pHPT, 20 matched healthy. All patients underwent surgery 1 to 3 months after the study.	Arterial diameters and thickness, blood pressure were not significantly different with respect to normal subjects and were unchanged 6 months after surgery.
Barletta [74]	2000, Italy	Single-blind, placebo-controlled, crossover study	Infusion of PTH	PWV IMT	5 healthy nonsmoker volunteers	There were no significant differences in basal echocardiographic measurements during PTH infusion with respect to placebo and in the hemodynamic response to tilt.
Kosch et al. [75]	2001, Germany	Prospective case–control study	pHPT	PWV IMT	20 patients assessed at baseline and 6 months after PTx and 20 healthy volunteers	No difference found at baseline and 6 months after PTx
Rubin et al. [66]	2005, USA	Cross-sectional case–control study	Mild pHPT	AIx	39 patients, 134 healthy subjects	AIx was also directly correlated with evidence of more active parathyroid disease, including higher PTH levels and lower bone mineral density
Tordjman et al. [78]	2010, Israel	Retrospective cohort study	Hypercalcemic (HC) and normocalcemic (NC) pHPT	PWV AIx	32 patients with NC-pHPT, 81 patients with mild HC-pHPT and a group of non-PHPT control subjects selected to match the patients’ population	CV or cerebrovascular disease was more common in the HC-PHPT group. Arterial stiffness parameters did not differ and were unrelated to serum calcium or PTH concentration
Rosa et al. [68]	2011, Czech Republic	Prospective case–control study	Hypertensive and normotensive patients with pHTP	PWV	28 patients with pHPT and concomitant hypertension,16 with pHTP without hypertension, 28 essential hypertensive patients and 18 healthy controls	PWV was significantly higher in patients with PH and hypertension when compared with patients with essential hypertension. Similarly, PWV was significantly higher in patients without hypertension in comparison with healthy controls. Specific treatment by PTX significantly decreases PWV, which may be determined primarily by improved BP control after surgery.
Schillaci et al. [67]	2011, Italy	Prospective case–control study	pHTP	PWV	24 patients with pHTP and 48 healthy controls; 17 patients underwent surgical PTx and were examined 4 weeks later.	Aortic PWV was significantly higher among pHTP patients. Aortic PWV decreased after surgery. The change in aortic PWV remained significant also after adjustment for changes in blood pressure
Ring et al. [76]	2012, Sweden	Prospective case–control study.	pHTP patients who underwent surgery	AIx aPWV rIMT cIMT	48 patients with mild PHPT without any known cardiovascular risk factors were studied at baseline and at one year after parathyroidectomy in comparison with 48 healthy age- and gender-matched controls.	Only aoPWV was slightly higher in patients than in the control group at baseline. PTx did not cause any change in indices of vascular function or arterial wall thickness
Stamatelopoulos et al. [73]	2014, Greece	Cross-sectional case–control study	pHPT and menopause	FMD PWV AIx IMT	102 postmenopausal women with pHPT and 102 women matched 1:1 for age and menopausal status, were consecutively recruited.	Women with pHPT had higher aortic and peripheral BP (*p* < 0.05 for all) but no correlation was observed with subclinical atherosclerosis
Cansu et al. [69]	2016, Turkey	Prospective case–control study	pHTP	AIx PWV IMT	16 normocalcaemic and 17 hypercalcaemic newly diagnosed asymptomatic PHPT patients and 15 age and body mass index (BMI) matched, healthy, normocalcaemic female control subjects; 17 hypercalcaemic patients who underwnt PTx	CIMT and PWV values in the HC and NC patients were higher than in the control group. There was a significant reduction in cIMT at the end of the 6th month after PTx,
Wetzel et al. [71]	2017, Germany	Cross-sectional data from the randomized, double-blind, placebo-controlled trial	pHPT	PWV	76 patients with treatment-naïve PTH levels.	PTH was independently associated with 24 h PWV.
Ejlsmark-Svensson et al. [77]	2019, Denmark	RCT	pHPT patients eligible for PTx.	PWV AIx	69 patients with PHPT; 33 underwent PTx, 36 were allocated in the control group	Changes in PWV, augmentation index and ambulatory 24 h BP did not differ between groups, except for an increase in ambulatory diastolic BP following PTX. However, in patients with baseline levels of ionized calcium > 1.45 mmol/L, PWV decreased significantly in response to PTX compared with the control group
Underbjerg et al. [82]	2019, Denmark	Cohort study	Ns-HypoPT and pseudohypoparathyroidism	PWV AIx	56 patients with Ns-HypoPT with 30 patients diagnosed with pseudohypoparathyroidism	PWV was significantly higher among patients with Ns-HypoPT, even after adjustment for mean arterial pressure, body mass index, age and gender.
Sumbul et al. [72]	2019, Turkey	Perspective study	pHPT	cIMT aIMT	65 patients and 30 healthy controls.	Aortic IMT is more useful than carotid IMT in showing vascular organ involvement in patients with primary hyperparathyroidism
Buyuksimsek et al. [70]	2020, Turkey	Cross-sectional study	Hypertension and pHTP	PWV	83 hypertensive patients with pHTP and 83 age and gender matched hypertensive controls	PWV significantly increases in newly diagnosed hypertensive patients with PHP and significantly related to serum calcium level.
Pamuk et al. [81]	2020, Turkey	Cross-sectional case–control study	Hypoparathyroidism	PWV	42 patients and 60 matched volunteers	PWV was found higher in the hypoparathyroidism group.

PTH: parathyroid hormone; pHPT: primary hyperparathyroidism; Ns-HypoPT: non-surgical hypoparathyroidism; PTx: parathyroidectomy; PWV: pulse wave velocity; IMT: intima-media thickness; AIx: augmentation index; cIMT: carotid intima-media thickness; aIMT: aortic intima-media thickness; rIMT: radial intima-media thickness; FMD: flow-mediated dilation; BP: blood pressure.

## 4. Discussion

### 4.1. Thyroid

Over recent years, the role of thyroid hormones in cardiovascular health has been widely studied. Several mechanisms of action of fT3, fT4 and TSH on arterial stiffness have been hypothesized. Thyroid hormones may affect the cardiovascular system by direct effects on arterial vessels (by regulating smooth muscle cells tone and endothelial function), on the heart (by influencing heart rate, rhythm, myocardial contraction and perfusion) or indirectly by influencing cardiovascular risk factors [7].

Focusing on effects on arterial vessels, thyroid hormones have both genomic and non-genomic mechanisms affecting vascular tone, by ion channel activation and regulation of specific signal transduction pathways.

Firstly, they have a cellular action on endothelium, causing production of nitric oxide via the phosphatidylinositol 3-kinase and serine/threonine protein kinase pathways [83]. Vasodilating effect on vascular smooth muscle cells and on resistance arteries results in widened pulse pressure and decreased systemic vascular resistance [84,85] in conditions of excess of thyroid hormones. Thyroid hormones thus lead to an increase in tissue oxygen consumption and in distending pressures. Higher mechanical stretch and altered perfusion patterns may lead to arterial vascular remodeling and to an increase in arterial, and in particular of aortic, stiffness [84,86].

Secondly, due to its positive inotropic and chronotropic effect, fT3 is directly responsible for acute hemodynamic changes [85], which affect the functional determinants of aortic stiffness [87]. The increase in heart rate and the increase in cardiac output driven by an excess of thyroid hormones lead to an altered hemodynamic adaptation. The resulting increase in mean arterial pressure and the shortening of left ventricular ejection may represent the major hemodynamic determinants of a functional increase in aortic stiffening. Additionally, an increase in heart rate worsens the pressure supply–demand balance, which may cause ischemia of the heart and of tissues with high blood flow supply. These conditions have been widely associated with aortic stiffening [88].

Thirdly, by acting on cardiovascular risk factors, thyroid hormones may indirectly lead to an increase in arterial stiffness. Notably the main determinant of aortic stiffness are blood pressure levels, which may be influenced by thyroid hormones. In hyperthyroidism, caused by inotropic effect on the heart and reduction in systemic vascular resistance, an increase in systolic and pulse pressure is often seen. Hypothyroidism is conversely associated with diastolic hypertension, induced by increased peripheral vascular resistance and by changes in circulating volume [89]. Variations in blood pressure components may acutely and chronically damage the arterial wall thus leading to arterial stiffness. Thyroid hormones are involved in lipid metabolism regulation. Hypothyroidism leads to an increased total cholesterol and LDL cholesterol [90], thus influencing atherosclerotic plaque burden. Non-HDL cholesterol is closely correlated with residual cardiovascular risk and arterial stiffness markers [91], although lipid levels are more linked with atherosclerotic vascular phenotypes, rather than to arteriosclerosis which is the hallmark of arterial stiffness. Thyroid hormones may be also linked to arteriosclerosis through different metabolic pathways: chronic inflammation, oxidative stress, insulin resistance [7].

An additional cause that may affect the promotion of cardiovascular disease in the presence of thyroid disease is the autoimmune process of disease per se [92,93]. Regardless of thyroid hormones level, autoimmune processes in the thyroid gland may cause a low-grade inflammation which plays a relevant role in the atherosclerotic process and in the stiffening of arteries [94]. Subclinical inflammation has been associated with arterial stiffness in the general population [95] with a causative role played by an increase in cytokine levels (Interleukin 1 and 6, tumor necrosis factor-β) and reactive oxygen species in the degradation of elastin, migration of smooth muscle cells and increase in collagen in the arterial wall.

Regarding the effect of treatment of thyroid and parathyroid diseases in relation to thyroid disease, most studies observed an improvement in vascular markers after correction of hormone levels. In hypothyroidism, an improvement in AIx (which is mainly a functional parameter) followed correction with hormone replacement therapy. In hyperthyroidism, treatment with antithyroid drugs may restore both AIx and PWV levels, although the evidence is scarce.

We should consider an important limitation in our work, which is not having considered articles not in the English language. Considering the important heterogeneity of studies considering effects of thyroid hormones on stiffness and the complex interactions of factors determining vascular dysfunction, a further contribution is given by a recent meta-analysis, which found that both hypothyroidism and thyrotoxicosis are associated with an increase of aortic stiffness [96]. The scheme represented in Figure 1 represents the impact of thyroid disease on arterial stiffness and the effects and mutual interactions of all the previously discussed factors.

### 4.2. Parathyroid

In pHPT, a prolonged exposure to high levels of PTH may affect the vascular system in different ways, leading to vascular functional and structural changes and to arterial stiffness. Arterial remodeling is induced by PTH with a few mechanisms. A direct effect of PTH on endothelial and vascular smooth muscle cells was hypothesized [97] and confirmed by evidence of a stimulatory effect on nitric oxide synthase, which may contribute to vascular injury and arteriosclerotic progression through reactive oxygen species [60]. PTH may induce expression of other mediators associated with adverse vascular remodeling, as interleukin-6, receptor for advanced glycation end-products and vascular endothelial growth factor [61,98]. An effect mediated by increased intracellular calcium influx was also suggested [99], due to altered calcium metabolism in vascular smooth muscle cells, which may lead to increased arterial resistances [62]. Furthermore, PTH may also have systemic actions, by inducing the activity of renin-angiotensin-aldosterone system as well as the sympathetic nervous system [100,101], which have been notably associated with adverse effects on arterial stiffness [4]. Chronically elevated levels of circulating calcium may also mediate the association between elevated PTH and arterial stiffness. The association of calcium levels with stiffness was also demonstrated at a population level [102] and most likely mediated by the induction of vascular calcifications [103].

An elevation of blood pressure levels, which is a hallmark of pHPT, is associated with functional changes in arterial viscoelastic properties of the large arteries and an increase in the blood pressure-dependent component of arterial stiffness [104]. In experimental conditions, the infusion of physiologic doses of PTH in otherwise healthy adults, is known to produce an increase in blood pressure [105]. The induction of a chronic hypertensive state may also lead to the adverse arterial remodeling typical of hypertension and to arterial stiffness [106].

Regarding possible therapeutic approaches, surgical treatment of pHPT may improve arterial stiffness [79], and thus produce a favorable effect on cardiovascular risk profile in these patients.

The factors and the mechanisms influencing arterial stiffness in pPHT are schematically represented in Figure 1.

## 5. Conclusions

Our review of clinical studies prompts that thyroid and parathyroid diseases are able to affect arterial stiffness biomarkers, thus leading to possible adverse cardiovascular outcomes. In hypothyroidism, most studies agree that an increase in arterial stiffness is present in both subclinical and overt hypothyroidism, although there is no complete agreement among methodologies evaluating the different aspects of vascular stiffening in the arterial vasculature. Levothyroxine replacement therapy in most studies has shown to lead to an improvement in arterial stiffness markers. Regarding hyperthyroidism, few studies observed an increase in biomarkers of structural arterial stiffening, although a relevant methodological heterogeneity prevents a definite conclusion. The increased heart rate typical of hyperthyroidism significantly alters the blood pressure profile, affecting biomarkers of functional arterial stiffness. In primary hyperparathyroidism, an increase in arterial stiffness is evident also in the mild forms. Arterial stiffening is reversed by parathyroidectomy, suggesting a role of surgery in reducing cardiovascular risk. Regarding methods used to quantify arterial stiffness, most studies focused on PWV, which represents the gold standard method to evaluate this parameter and may thus represent the preferred method in the vascular evaluation of thyroid and parathyroid diseases.

Thyroid and parathyroid diseases are systemic diseases, characterized by an increase of cardiovascular risk, which present an increase in stiffness of the large arteries. Arterial stiffness measurement can be effectively used in the clinical evaluation of these conditions in order to quantify and possibly reduce the risk of cardiovascular events.

## Figures and Tables

**Figure 1 jcm-11-03146-f001:**
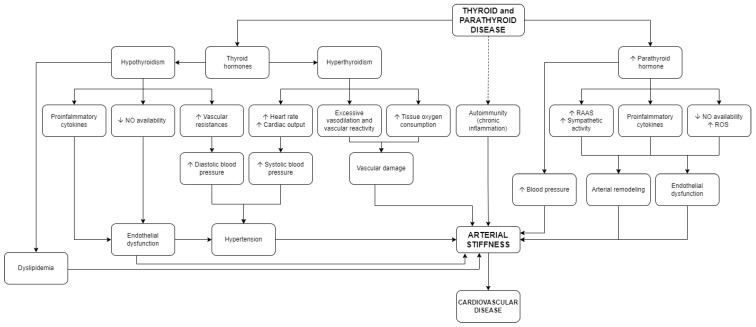
Pathophysiology of arterial stiffness in thyroid and parathyroid disease. RAAS: renin-angiotensin-aldosterone system. NO: nitric oxide. ROS: reactive oxygen species.

## Data Availability

Not applicable.

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
