# Peer review of "Arterial Stiffness in Thyroid and Parathyroid Disease: A Review of Clinical Studies"

_jcm, 2022, doi:10.3390/jcm11113146_

Round 1
Reviewer 1 Report
The topic of this paper regarding Arterial stiffness in thyroid and parathyroid disease is interesting. The paper is generally well-written and seems to share comprehensive information on the topic with respect to the current literature. I have some remarks:
Minor comments:
The title of the abstract sections should be removed (Background, Methods, Results, etc.).
In part Introduction section, the authors should more specifically define the role of hyperthyroidism and hypothyroidism in the development of cardiovascular risks (lines 44 to 51).
Once PWV is defined, you should use an abbreviation (for instance, lines 114, 115).
Lines 168 to 175 – Authors should check the location of these abbreviations. The list of abbreviations may be at the end of the manuscript.
Line 193, the authors should be more specific about which conflicting study data they found.
All abbreviations in Figure 1 should be explained below the Figure (for instance: ROS).
Abbreviations in the Tables should be explained below the Tables.
In Table 2, the author was able to add the name of disease (for instance: column – disease in last row – overt hyperthyroidism (Grave´s disease)).
Author Response
We thank the reviewers for their thorough comments that permitted to improve our manuscript.
We believe that we fully answered to their comments in this reviewed version.
We reported in this document the answers to their comments along with the changes we have made in the text.
REVIEWER 1
The title of the abstract sections should be removed (Background, Methods, Results, etc.).
We removed the titles of the abstract sections, as requested.
In part Introduction section, the authors should more specifically define the role of hyperthyroidism and hypothyroidism in the development of cardiovascular risks (lines 44 to 51).
We added more detailed information on how cardiovascular risk factors influence the risk of cardiovascular events in hypothyroidism and hyperthyroidism. Further mechanisms influencing cardiovascular risk in these diseases are provided all along the paper, in the Results and Discussion sections.
Introduction, Paragraph 2: “Patients with overt hyperthyroidism or hypothyroidism show several alterations, caused either by effects of thyroid hormones in the heart and in the vasculature or by cardiovascular risk factors (including blood pressure, dyslipidemia and inflammation), which may increase cardiovascular risk and lead to cardiovascular morbidity and mortality”
Once PWV is defined, you should use an abbreviation (for instance, lines 114, 115).
We corrected using the abbreviation in lines 114-115.
Lines 168 to 175 – Authors should check the location of these abbreviations. The list of abbreviations may be at the end of the manuscript.
Thank you for highlighting this issue. We corrected the list of abbreviations and placed it at the end of the manuscript.
Line 193, the authors should be more specific about which conflicting study data they found.
Thank you. We corrected as follows:
“By contrast, the studies evaluating the effects of parathyroidectomy upon arterial stiffness have reported conflicting data regarding effects on arterial structure and function”
All abbreviations in Figure 1 should be explained below the Figure (for instance: ROS).
We added the list of abbreviations below the Figure 1.
Abbreviations in the Tables should be explained below the Tables.
We added the list of abbreviations below the Tables.
In Table 2, the author was able to add the name of disease (for instance: column – disease in last row – overt hyperthyroidism (Grave´s disease)).
Thank you. We showed the name of the disease in the “Disease” column in Table 2.

Reviewer 2 Report
This article addresses a clinically relevant issue that justifies this systematic literature review or descriptive review.
It is formally correct.
However, we have some questions:
- Considering only the English language is a limitation, because there are, for example, many articles in French in this area.
- Tha authors do not refer if they consider only open or free articles. The criteria used in the selection of articles must be clearly described.
- It would be useful to have a description of the parameters considered to define the PWV, for example.
- Perhaps a framework or tale taking into account only the PWV would be useful, because it is the gold standard for arterial stiffness.
- There could be a definition of what is considered subclinical hyperthyroidism or subclinical parathyroidism.
- There should be a better discussion of arterial stiffness in relation to calcium and lipid levels.
-In the discussion there should be more emphasis on the relationship of thyroid and parathyroid hormones with metabolism, as thyroid hormones are very important in this aspect. There should also be greater emphasis on modifying arterial stiffness with surgical or medical treatment of thyroid and parathyroid diseases (greater emphasis and description of what is already done in the article).
In conclusion, it should include, at least in the discussion, some of the aspects that we have highlighted.
Author Response
ANSWER TO REVIEWERS
Manuscript title: Arterial stiffness in thyroid and parathyroid disease: a review of clinical studies
We thank the reviewers for their thorough comments that permitted to improve our manuscript.
We believe that we fully answered to their comments in this reviewed version.
We reported in this document the answers to their comments along with the changes we have made in the text.
REVIEWER 2
Considering only the English language is a limitation, because there are, for example, many articles in French in this area.
Thank you for highlighting this limitation. We included the following sentence in the Discussion:
Discussion, “Thyroid” section, last paragraph: “We should consider an important limitation in our work, which is not having considered articles not in the English language.”
Tha authors do not refer if they consider only open or free articles. The criteria used in the selection of articles must be clearly described.
We considered all articles and not only open or free articles. We clarified this in “Methods” section:
Methods: “We considered all papers in open-access and non-open access journals.”
It would be useful to have a description of the parameters considered to define the PWV, for example.
We added a definition of PWV in the Results section:
Results, Paragraph 1: PWV is the velocity at which the pressure pulsation propagates through the circulatory system, and is defined as the distance travelled by the pulse wave divided by the time [2], and the carotid-femoral PWV is the gold standard for measuring stiffness of the aorta [3].
Perhaps a framework or tale taking into account only the PWV would be useful, because it is the gold standard for arterial stiffness.
We agree with the reviewer that PWV, representing the gold standard measure for arterial stiffness, is the more suitable method for evaluating this parameter in thyroid and parathyroid disease. The importance of PWV is highlighted all along the paper, and the majority of references is based on studies using PWV. We added a sentence in the Conclusions regarding the possibility to use PWV in the evaluation of arterial stiffness in these diseases.
Conclusions, Paragraph 1: “Regarding methods used to quantify arterial stiffness, most studies focused on PWV, which represents the gold standard method to evaluate this parameter and may thus represent the preferred method in the vascular evaluation of thyroid and parathyroid diseases.”
There could be a definition of what is considered subclinical hyperthyroidism or subclinical parathyroidism.
We defined the various conditions in the text:
Results, section “Overt and subclinical hypothyroidism”, Paragraph 1: “subclinical hypothyroidism, defined as a serum thyroid stimulating hormone (TSH) level above the upper normal limit despite normal levels of serum free thyroxine (fT4)”.
Results, section “Overt and subclinical hyperthyroidism”, Paragraph 1: “subclinical hyperthyroidism (defined as a low or undectable TSH, with normal fT3 and fT4)”
Results, section “Parathyroid”, Paragraph 1: “mild pHPT, who exhibit inappropriately high levels of parathyroid hormone (PTH) and normal or only mildly elevated calcium levels”.
There should be a better discussion of arterial stiffness in relation to calcium and lipid levels.
We thank the reviewer for the suggestion. We added the following sentences to the Discussion:
Discussion, section “Thyroid”, Paragraph 5: “Hypothyroidism leads to an increased total cholesterol and LDL cholesterol [90], thus in-fluencing atherosclerotic plaque burden. Non-HDL cholesterol is closely correlated with residual cardiovascular risk and arterial stiffness markers [91], although lipid levels are more related with atherosclerotic vascular phenotypes, rather than to arteriosclerosis which is the hallmark of arterial stiffness”.
Discussion, section “Parathyroid”, Paragraph 5: “Chronically elevated levels of circulating calcium may also mediate the association be-tween elevated PTH and arterial stiffness. The association of calcium levels with stiffness was also demonstrated at a population level [102] and most likely mediated by the induc-tion of vascular calcifications [103].”
In the discussion there should be more emphasis on the relationship of thyroid and parathyroid hormones with metabolism, as thyroid hormones are very important in this aspect. There should also be greater emphasis on modifying arterial stiffness
